# Roles of the PARP Inhibitor in *BRCA1* and *BRCA2* Pathogenic Mutated Metastatic Prostate Cancer: Direct Functions and Modification of the Tumor Microenvironment

**DOI:** 10.3390/cancers15092662

**Published:** 2023-05-08

**Authors:** Takahiro Inoue, Sho Sekito, Takumi Kageyama, Yusuke Sugino, Takeshi Sasaki

**Affiliations:** Department of Nephro-Urologic Surgery and Andrology, Mie University Graduate School of Medicine, 2-174 Edobashi, Tsu 514-8507, Japan; momosekisho@gmail.com (S.S.); kagetaku@med.mie-u.ac.jp (T.K.); y-sugino@med.mie-u.ac.jp (Y.S.); t-sasaki@med.mie-u.ac.jp (T.S.)

**Keywords:** prostate cancer, *BRCA1/2*, PARP inhibitor, tumor microenvironment, DNA damage repair

## Abstract

**Simple Summary:**

Recent genomic analytical advancements have revealed that *BRCA1/2* pathogenic variants are the most frequent mutations among DNA damage repair genes in prostate cancer. Polyadenosine diphosphatase ribose polymerase (PARP) inhibitor, olaparib, has been shown as an effective therapeutic option for the disease. This review focuses on PARP inhibitors’ basic and clinical mechanisms of action against prostate cancer and discusses their effects on the tumor microenvironment.

**Abstract:**

Cancer cells frequently exhibit defects in DNA damage repair (DDR), leading to genomic instability. Mutations in DDR genes or epigenetic alterations leading to the downregulation of DDR genes can result in increased dependency on other DDR pathways. Therefore, DDR pathways could be a treatment target for various cancers. In fact, polyadenosine diphosphatase ribose polymerase (PARP) inhibitors, such as olaparib (Lynparza^®^), have shown remarkable therapeutic efficacy against *BRCA1/2*-mutant cancers through synthetic lethality. Recent genomic analytical advancements have revealed that *BRCA1/BRCA2* pathogenic variants are the most frequent mutations among DDR genes in prostate cancer. Currently, the PROfound randomized controlled trial is investigating the efficacy of a PARP inhibitor, olaparib (Lynparza^®^), in patients with metastatic castration-resistant prostate cancer (mCRPC). The efficacy of the drug is promising, especially in patients with *BRCA1/BRCA2* pathogenic variants, even if they are in the advanced stage of the disease. However, olaparib (Lynparza^®^) is not effective in all *BRCA1/2* mutant prostate cancer patients and inactivation of DDR genes elicits genomic instability, leading to alterations in multiple genes, which eventually leads to drug resistance. In this review, we summarize PARP inhibitors’ basic and clinical mechanisms of action against prostate cancer cells and discuss their effects on the tumor microenvironment.

## 1. Introduction

The standard medical treatment for advanced prostate cancer involves medical or surgical castration. This is significantly different from other cancer types, such as breast and lung cancers, for which treatment is selected based on molecular mechanisms [1,2]. Recently, it has become a standard treatment to use androgen receptor signaling inhibitors (ARSIs) or/and docetaxel in addition to medical or surgical castration from the initial treatment, according to the risk of prostate cancer [3]. Olaparib (Lynparza^®^) is the first drug based on gene mutations, especially *BRCA1* and *BRCA2* pathogenic mutations, for the treatment of advanced prostate cancer [4] and was highly expected to mark the dawn of personalized medicine in this field.

DNA is constantly under repair because it is damaged by products of cellular metabolism, exposure to environmental agents, and chemical bonds that spontaneously disintegrate under physiological conditions [5]. Normal and cancerous cells depend on multiple DNA damage response pathways to repair different forms of DNA damage [6]. The most common form of DNA damage is a single-strand discontinuity, known as a single-strand break (SSB), which is predominantly repaired through base excision repair mechanisms but also by nucleotide excision repair and, to a lesser extent, by DNA mismatch repair [5]. Upon DNA SSB damage, polyadenosine diphosphatase ribose polymerase (PARP), such as PARP1, sensitizes the site of SSBs for DNA repair, and SSBs are repaired by the orchestration of various proteins. If the repair of SSBs is deficient or disabled, they can be converted into double-strand breaks (DSBs) owing to the collapse or blockage of DNA replication forks during the S phase of the cell cycle. The main DSB repairs are error-free homologous recombination (HR) and error-prone non-homologous end joining (NHEJ) [7].

Cancer cells frequently have defects in DNA damage repair (DDR), leading to genomic instability. Mutations in DDR genes or epigenetic alterations leading to the downregulation of DDR genes can result in increased dependency on other DDR pathways; therefore, DDR pathways could be a treatment target. For example, PARP inhibitors such as olaparib (Lynparza^®^) have shown remarkable therapeutic efficacy against *BRCA1/2*-mutant cancers through synthetic lethality [4].

In this review, we discuss the fundamental mechanisms underlying the efficacy of PARP inhibitors against DDR pathways in prostate cancer cells, the clinical aspects of PARP inhibitors in prostate cancer, and their role in the tumor microenvironment.

## 2. Adenosine Diphosphate-Ribosylation

In adenosine diphosphate (ADP)-ribosylation, nicotinamide adenine dinucleotide (NAD+) is cleaved into nicotinamide and ADP-ribose, which are then transferred onto the substrate. The main targets are proteins; however, DNA, RNA, and their metabolites have also been shown to undergo ADP-ribosylation [8,9]. ADP-ribosylation is involved in various biological processes, including DNA-damage repair, chromatin and transcriptional regulation, cellular senescence, apoptosis, and immunity. ADP-ribosyltransferase (ART) is mainly responsible for ADP-ribosylation and is based on three conserved amino acids in the catalytic domain of NAD+. ARTs are classified into diphtheria toxin-like ART (ARTD) and cholera toxin-like ART (ARTC), with most mammalian ARTs belonging to ARTD. The human poly (ADP-ribosyl) polymerase (PARP) also belongs to the ARTD family, and PARPs involved in DNA damage repair include PARP-1, PARP-2, and PARP-3 [9,10]. PARP-1 is the major DNA damage repair PARP in terms of its abundance and enzymatic output in response to DNA damage. Dysregulation of ADP-ribosylation causes various diseases, such as neurodegenerative diseases, cancer, and inflammation [9].

## 3. Poly (ADP-Ribosyl) Polymerase-1 (PARP-1)

PARP-1 is one of the main PARPs involved in repair after DNA damage and is composed of seven domains arranged linearly in a bead-like structure. When an SSB occurs, it is immediately recognized and transformed into an active conformation along with DNA binding. NAD+ is more likely to bind to the active site of PARP-1, and the activity of PARP-1 is further enhanced by NAD+ binding. During this process, multiple ADP-ribose binding (PARylation) to PARP-1 and PARylation of histones around the SSB occur. Additionally, self-PARylated PARP-1 is thought to dissociate from DNA, which is negatively charged, as the ribosyl polymer becomes negatively charged. Through this dissociation, XRCC1, APTX, PNKP, DNA polymerase b, and ligase 3a are recruited to the SSB site and SSB repair occurs [10].

## 4. PARP Inhibitors

Talazoparib, olaparib, veliparib, niraparib, rucaparib, fuzuloparib, and pamiparib are among the PARP inhibitors (PARPi) that are under development or already in clinical use in both Japan and abroad [5]. Its mechanism of action involves the inhibition of NAD+ binding to the PARP active site, which is common to all drugs, resulting in self-PARylation inhibition. It has been proposed that conformational changes alter the DNA binding state [9]. Moreover, the strength of the inhibition differs depending on the drug, with talazoparib being the strongest inhibitor and veliparib being the weakest, with the degree affecting the difference in adverse events [11]. Tissue distribution and efficacy was also different among these PARPi; niraparib showed highest tumor and bone marrow distribution [12]. Typically, homologous recombination repair (HRR) is performed together with *BRCA1*, *BRCA2*, and others, and DSBs repair is performed accurately [7]. However, when SSBs cannot be repaired using PARP inhibitors, their accumulation causes replication fork collapse and double-strand breaks (DSBs) during DNA replication [7]. This results in cell-cycle arrest and synthetic lethality [13,14,15].

## 5. Characteristics of Prostate Cancer in Cases with *BRCA1/BRCA2* Pathogenic Variants

HRR is a representative example of the DDR mechanism, and *BRCA1/BRCA2*, *ATM*, and *CDK12* are well-known genes responsible for this double-strand repair (HRR-related genes) [7]. It has been reported that approximately 12% of patients with metastatic prostate cancer have pathogenic gene variants related to the DDR mechanism in their germline [16]. *BRCA1/BRCA2* pathogenic variants are the most frequent mutations among DDR genes, with a reported frequency of approximately 6% [16].

In prostate cancer, the frequencies of germline and somatic variants of *BRCA1/BRCA2* are almost the same; therefore, evaluation of both germline and somatic variants is necessary [17]. This is different from ovarian or breast cancers and should be considered accordingly [17]. *BRCA1/BRCA2* pathogenic variants are reported to occur in 4–18% of patients with localized/castration-resistant prostate cancer in both germline and somatic cell lineages [17,18].

A prospective observational research report on the incidence of prostate cancer in western populations focused on a group of *BRCA1/BRCA2* pathogenic variant carriers and a control group that underwent predictive testing and finally tested negative for a pathogenic germline *BRCA1/BRCA2* variant [19]. As a result of a three-year screening, patients with a *BRCA2* pathogenic variant had a higher incidence of prostate cancer, a higher proportion of clinically significant cancers, and a lower age of onset [19]. In particular, the number of cancers with a Gleason score of seven or higher tends to be higher in cases with *BRCA2* pathogenic variants than in those without variants [19]. However, no such trend has been observed in the carriers of *BRCA1* pathogenic variants [19]. In addition, prospective observation of *BRCA1* and *BRCA2* pathogenic variant carriers in the United Kingdom and Ireland (median observation period of 5.9 years and 5.3 years, respectively) showed that those with *BRCA2* pathogenic variants showed a two to five times higher incidence of prostate cancer than that of the general population [20]. Furthermore, the prevalence of prostate cancer is much higher in patients with a family history of a *BRCA2* pathogenic variant [20]. However, this report did not find a tendency for the incidence of prostate cancer to be higher in carriers of *BRCA1* pathogenic variants than in the general population [20].

In an analysis of germline mutations in Japan, 1.1% of prostate cancer cases were found to have *BRCA2* pathogenic variants (0.2% in the control group) [21]. Moreover, frequencies of other germline mutations were detected in the *HOXB13* (0.8% in the prostate cancer cases and 0.2% in the control group), *ATM* (0.5% in the prostate cancer cases and 0.2% in the control group), and *BRCA1* (0.2% in the prostate cancer cases and 0.1% in the control group) genes, in this order [21]. It has also been reported that 24.5% of men with a *BRCA2* germline pathogenic variant develop prostate cancer by the age of 85 [22].

In a prospective cohort study of 419 patients with metastatic castration-resistant prostate cancer (68 harboring germline pathogenic variants in genes involved in DNA damage repair mechanisms, including 14 with *BRCA2*, 8 with *ATM*, and 4 with *BRCA1*), cancer-specific survival rates were significantly lower in individuals with a germline *BRCA2* pathogenic variant than in individuals without germline pathogenic variants in genes involved in DNA damage repair mechanisms [23].

In addition, among localized prostate cancers under active surveillance in the United States, prostate cancer patients with germline *BRCA1/BRCA2* or *ATM* pathogenic variants were 1.96 times more likely to be upgraded on re-biopsy specimens than prostate cancer patients without pathogenic variants [24]. In particular, prostate cancer patients with a *BRCA2* pathogenic variant who were on active surveillance were 2.74 times more likely to have an upgraded Gleason score [24]. Therefore, patients with prostate cancer and pathogenic variants of *BRCA1/BRCA2* or *ATM* are not appropriate candidates for active surveillance, even if other clinical and pathological features are acceptable.

## 6. The PROfound Study and Olaparib (Lynparza^®^)

The PROfound study is a randomized prospective Phase 3 controlled trial investigating the efficacy of the PARP inhibitor, olaparib (Lynparza^®^), in patients with metastatic castration-resistant prostate cancer (mCRPC) after progression on enzalutamide or abiraterone treatment [4]. These patients had pathogenic variants in genes associated with DNA repair pathways in tumor tissue based on the FoundationOne CDx cancer genomic profile [4]. Cohort A included mCRPC cases with *BRCA1*, *BRCA2*, or *ATM* variants, whereas Cohort B included patients with mutations in any of the other 12 genes involved in DNA repair. In Cohort A, 162 patients received 600 mg olaparib (Lynparza^®^) and 83 patients received 160 mg enzalutamide or 1000 mg abiraterone plus 10 mg predonisone. In contrast, in Cohort B, 94 patients received olaparib (Lynparza^®^) and 48 patients received enzalutamide/abiraterone. Image-based progression-free survival (rPFS) in Cohort A was 7.4 months for olaparib (Lynparza^®^) and 3.6 months for the enzalutamide/abiraterone group, with a hazard ratio of 0.34, *p* < 0.001, and significantly higher rPFS in the olaparib (Lyparza) group [4]. In addition, overall survival (OS) was 19.1 months for the olaparib (Lynparza^®^) group and 14.7 months for the enzalutamide/abiraterone group in Cohort A, with a hazard ratio of 0.69 and *p* = 0.02, indicating a significant prolongation of OS in the olaparib (Lynparza^®^) group. However, in Cohort B, OS was 14.1 months for the olaparib (Lynparza^®^) group and 11.5 months for the enzalutamide/abiraterone group, which were not significantly different [25].

The above results demonstrated the usefulness of olaparib (Lynparza^®^) in reducing mortality for mCRPC with any of the *BRCA1*, *BRCA2*, or *ATM* gene variants (pathogenic variants), compared to enzalutamide/abiraterone. However, due to a retrospective sub-analysis, its usefulness was not recognized in patients with *ATM* gene variants; therefore, in Japan, olaparib (Lynparza^®^) is covered by health insurance only for mCRPC with *BRCA1* and *BRCA2* pathogenic variants [26].

## 7. Olaparib (Lynparza^®^) and Genetic Testing

In February 2023 in Japan, companion diagnostic tests for determining the indications for olaparib, FoundationOne^®^ CDx (F1CDx), FoundationOne^®^ Liquid CDx (F1LiquidCDx), and BRACAnalysis diagnostic systems were approved by the government and covered by public insurance if patients become refractory against an ARSI [26]. The recommended timeline for a molecular test is different between Japan and US/European countries; in the latter countries, it is recommended when diagnosing metastatic disease [3,27]. F1CDx and F1LiquidCDx detect somatic and germline mutations indiscriminately; therefore, germline origin cannot be determined [28]. In contrast, BRACAnalysis can detect only germline variants [29]. As mentioned above, approximately 50% of *BRCA1/2* mutations in prostate cancer are somatic lineage mutations; therefore, it is important to consider this when selecting diagnostic methods. Germline mutations have also been reported in older-onset prostate cancer [21]. Moreover, in metastatic prostate cancer, there is no difference in the frequency of DDR gene mutations depending on the family history and age of onset [16]. Therefore, no scientific basis for narrowing down the subjects of this test based on family history and age was reported [16,21].

F1CDx and F1LiquidCDx also function as cancer gene panel tests, and the OncoGuide NCC Oncopanel System has the same function in Japan [30]. However, while the OncoGuide NCC Oncopanel System has a problem in that the androgen receptor, which is the most frequent genetic mutation in mCRPC, is not included in the gene list to be tested [30], it is possible to perform a gene panel test using blood and tissue [30]. Therefore, it has the advantage of simultaneously discriminating between variants derived from somatic and germline mutations [30]. F1CDx detects mutations such as nucleotide substitutions, insertions/deletions, copy number abnormalities, gene rearrangements, and homozygous deletions in 324 genes and can be used to determine the microsatellite instability (MSI) of tumors and calculate the gene mutation burden (tumor mutational burden: TMB) [31]. Furthermore, F1LiquidCDx can detect nucleotide substitutions, insertions/deletions, copy number abnormalities, and gene rearrangements in the same 324 genes as F1CDx; however, it has not been approved by insurance for determining homozygous deletions, MSI, or calculating TMB, at least in Japan. At present, to implement F1LiquidCDx, there is a need for medical reasons such as difficulties in performing tests on tumor tissue specimens. Additionally, the concordance rate between tissue and circulating tumor DNA results for *BRCA1/BRCA2/ATM* mutations in the PROfound test depends on the type of gene mutation (nonsense mutations and frameshift mutations have high concordance rates, gene rearrangements, and poor concordance for conjugative loss) and is generally favorable [28].

## 8. Background of Successful Cases of Olaparib (Lynparza^®^)

The efficacy of olaparib (Lynparza^®^) has been verified for mCRPC with pathogenic variants in *BRCA1* and *BRCA2*, but it is not effective in all cases [4]. A retrospective analysis of 123 patients with *BRCA1* and *BRCA2* pathogenic variants (*BRCA1*, 13 cases; *BRCA2*, 110 cases) who received PARP inhibitors revealed that patients with *BRCA2* pathogenic variants were more susceptible to olaparib than those with *BRCA1* variants and had longer PSA progression-free survival, progression-free survival, and overall survival [28]. This is partly due to the fact that more cases with a *BRCA1* pathogenic variant have a monoallelic mutation (many cases with a *BRCA2* pathogenic variant have a biallelic mutation). Among *BRCA1* and *BRCA2* pathogenic variants, cases with truncated mutations responded better to PARP inhibitors than those with missense mutations. In addition, patients without *TP53* mutations also respond better to PARP inhibitors than those with *TP53* mutations [32]. Genetic analysis performed in TOPARP-B, a Phase 2 trial of olaparib (Lyparza^®^) in mCRPC, showed similar effects of olaparib (Lyparza^®^) in *BRCA1/2* somatic and germline mutations [33]. However, among them, cases with *BRCA2* homozygous deletion showed the greatest response to olaparib (Lynparza^®^) [33]. In addition, although the number of cases was small, there were cases of fair responders who had biallelic mutations in the *PALB2* allele and *ATM* mutations with low protein expression by immunohistochemistry [33].

## 9. Mechanisms of Resistance to PARP Inhibitors

Reversion mutations that restore open reading frames are the most common cause of PARP inhibitor resistance in clinical settings [34]. This adaptation is often a partial reading frame restoration by the second gene mutation in the mutant *BRCA* gene, sufficient to sustain some of the key *BRCA1/2* activities (reversion mutation), resulting in the recapitulation of the HRR mechanism [34]. Consequently, the effects of PARP inhibitors can be attenuated [34].

Rucaparib, another PARP inhibitor (rucaparib is not covered by health insurance for prostate cancer in Japan but is approved by the FDA in the U.S.), was evaluated in the TRIRON2 trial, a phase two study targeting mCRPC cases with mutations in DNA repair pathway-related genes [34]. The trial focused on mCRPC cases that progressed after treatment with one or two new hormonal agents and taxane chemotherapy. In this trial, 39/100 patients had *BRCA* reversions, and the mutation frequency was the same regardless of the difference between *BRCA1* and *BRCA2* mutations and whether the underlying mutation was a somatic or germline mutation. Furthermore, the frequency tended to be higher in patients with high circulating DNA levels, and the presence or absence of reversion was not associated with any clinicopathological factors. Moreover, two or more reversion mutations were detected in 74% of the cases with reversion; however, the gene mutation frequencies detected in cell-free DNA were lower than the original *BRCA1/BRCA2* mutation frequencies in all cases [34]. The longer the effect of rucaparib, the more likely that these reversion mutations occurred, which is theoretically understandable considering the time of exposure to rucaparib.

In sporadic triple-negative breast tumors, a frequently detected mechanism of *BRCA1* inactivation is *BRCA1* promoter hypermethylation, resulting in its silencing [35]. Silencing is reversed by demethylation, which enables residual transcription and results in resistance to PARP inhibitors [35]. A previous study has shown the *BRCA1* gene under the control of a heterogeneous active promoter by chromosomal rearrangement, even though the gene promoter was hypermethylated [35]. Specific to *BRCA1*-mutated tumors, abrogation of 53BP1 and its interacting partners RIF1 and the shieldin complex reactivates end resection and loads RAD51 at DSB DNA sites, sufficient to bypass the HR function of *BRCA1* and confer PARP inhibitor resistance [36]. Furthermore, PARP phosphorylation by c-Met [37], PARP1 point mutations that interfere with the PARP1 DNA-binding zinc-finger domain [38], upregulation of P-glycoprotein [39], and the inactivation or deficiency of PAR glycohydrase (PARG) [40] have all been reported to be involved in PARP inhibitor resistance.

Various resistance mechanisms, other than *BRCA1/BRCA2* reversion mutations, have been reported in ovarian and breast cancer [5]; however, there have been few reports on prostate cancer. A recent report evaluated gene expression profiles between olaparib (Lynparza^®^)-resistant prostate cancer cell lines established after long-term exposure to the drug and the parent lines [41]. However, the cell lines used did not harbor *BRCA1/BRCA2* mutations; therefore, they did not necessarily reflect clinical resistance to PARP inhibitors [41]. Therefore, further studies regarding this are required.

## 10. Modulation of the Tumor Microenvironment by PARP Inhibitors

In addition to the tumor mutation burden (TMB), non-neoantigen-based mechanisms of tumor cell recognition by the host immunological system have been reported [42]. Cyclic GMP-AMP (cGAMP) synthase (cGAS) is a cytosolic DNA sensor that activates the innate immune response through the production of the second messenger cGAMP, activating the adaptor stimulator of the interferon (IFN) gene (STING) and its downstream pathway [42]. The cGAS-STING pathway not only mediates the protective immune system against infection by various DNA-containing pathogens but also detects tumor-derived DNA, which activates intrinsic antitumor immunity [42]. The PARP inhibitor olaparib induces CD8+ T cell infiltration and activation in vivo through activation of the cGAS/STING pathway in tumor cells with paracrine activation of dendritic cells and was more significant in HR-deficient than in HR-proficient breast cancer cells [43]. Moreover, PARP inhibitors promote the accumulation of cytosolic DNA fragments due to unresolved DNA lesions, activating the cGAS/STING pathway and stimulating the production of type I IFNs to induce antitumor immunity independent of *BRCA* mutations [44]. Upon treatment with the PARP inhibitor olaparib, increased expression of IFNb, a type I IFN, leads to the upregulation of chemokine expression, including CCL5 and CXCL10, PD-L1 expression, and T cell recruitment in small cell lung cancer cells [45]. Additionally, a PARP inhibitor such as olaparib (Lynparza^®^) upregulates PD-L1 expression primarily through GSK3β inactivation [46], indicating that PARP inhibitors render cancer cells more resistant against T-cell-mediated cell death [46]. These effects are observed in rucaparib-, olaparib-, and talazoparib-treated cancer models [46]. Furthermore, treatment with the PARP inhibitor olaparib upregulates PD-L1 expression through the formation of cytotoxic DSBs and activation of ATM/ATR/Chk1 kinases in *BRCA2*-depleted cancer cells independent of the IFN pathway [47]. Interestingly, cancer cell-specific genetic knockout of PARP1 in non-small cell lung cancer mouse models induces T lymphocyte-mediated tumor growth control coupled with signs of T cell activation in the local tumor microenvironment, suggesting that therapeutic PARP1 inhibitors might modulate the tumor microenvironment [48]. Based on these biological findings, a combination of PARP inhibitors and immune checkpoint inhibitors (ICIs) is currently under clinical testing.

In breast cancer cells, mutations in *BRCA1* and *BRCA2* differentially modulate the tumor-immune microenvironment, which may be partially due to distinct mutational and copy number profiles, resulting in differential responses to checkpoint blockade immunotherapy [49]. This suggests that *BRCA2* deficiency is associated with increased immunogenicity and an improved response to checkpoint blockade immunotherapy compared to *BRCA1* deficiency [49]. Furthermore, multiplex immunohistochemistry of T and B cells and quantitative spatial analysis of both germline HRD (gHRD) and sporadic prostate cancer showed that gHRD prostate cancer had a more T cell-inflamed microenvironment than sporadic tumors [50]. Further spatial tumor-immune microenvironment analysis by single-cell RNA sequencing in prostate cancer tissues may reveal more precise mechanisms of DDR defects in tumor and antitumor immunity.

PARP inhibitors induce CCL5 secretion via the NF-κB pathway in stromal fibroblasts, which in turn activate the fibroblasts in an autocrine manner [51]. Moreover, PARP inhibitors downregulate p62 by impairing PARylation of AP-1 transcription factors, resulting in cancer-associated fibroblast activation [52]. Therefore, we need to understand the mechanisms underlying the modulation of tumor-associated fibroblasts by PARP inhibitors to potentially enhance their effects against tumors (Figure 1).

The intracellular cyclic GMP-AMP (cGAMP) synthase (cGAS) stimulator of the interferon gene (STING) pathway senses tumor-associated DNA. Tumor cells that are killed by polyadenosine diphosphatase ribose polymerase (PARP) inhibitors are taken up by phagocytes, providing fragmented double-stranded DNAs. Cystosolic DNA binds to and activates cGAS, which catalyzes the synthesis of cGAMP from adenosine triphosphate (ATP) and GTP. cGAMP binds to the endoplasmic reticulum membrane adaptor STING, resulting in conformational changes to the active STING. Activated STING binds to the kinase TBK1, which in turn phosphorylates the transcriptional molecule IRF3. STING also activates the kinase IKK, which phosphorylates the IκB family of inhibitors of the transcription factor NF-κB. The dimerized phosphorylated IRF3 and NF-κB enters the nucleus and functions together to induce the expression of interferons and inflammatory cytokines. Interferons (IFNs) stimulate the maturation of phagocytes such as dendritic cells (DCs) and facilitate the presentation of tumor-associated antigens on the major histocompatibility complex I (MHC I) DCs migrate to the lymph nodes to activate CD8+ T cells, which finally attack the tumors. In contrast, PARP inhibitors upregulate PD-L1 expression through various molecular pathways, leading to T cell exhaustion. Moreover, PARP inhibitors induce CCL5 secretion via the NF-κB pathway in stromal fibroblasts, which in turn activate the fibroblasts in an autocrine manner. PARP inhibitors downregulate p62 by impairing the PARylation of AP-1 transcription factors, resulting in cancer-associated fibroblast activation. Thus, PARP inhibitors are the so-called “double-edged sword” for cancer treatment, and combination treatments together with PARP inhibitors could potentially enhance their effects against tumors. ER, Endplasmic reticulum; ERGIC, ER-Golgi intermediate compartment; IFNAR, Interferon-alpha receptor; IRF; Interferon regulatory factor; TCR, T-cell receptor; TKB1, TANK-binding kinase 1.

## 11. Future Application of PARP Inhibitors for Prostate Cancer

PARP1 activity is significantly increased in CRPC cells compared to that in hormone-sensitive prostate cancer cells, and PARP inhibitors cause the depletion of both the androgen receptor (AR) and PARP1 on DNA in CRPC cells in vitro and in vivo [53]. Moreover, AR can induce DSB repair in prostate cancer cells [53], which prompted us to move on to a combination treatment with a PARP inhibitor and androgen receptor signaling inhibitors (ARSIs) for patients with CRPC. A double-blind, phase three trial of abiraterone and olaparib (Lynparza^®^) versus abiraterone and a placebo in patients with mCRPC who had not received ARAT agents in the first-line setting was conducted (PROpel study). Patients were enrolled regardless of their HRR gene mutation status [54]. Median imaging-based progression-free survival (ibPFS), which was the primary endpoint, was 24.8 months in the combination group and 16.6 months in the abiraterone alone group, which was significantly longer in the former (hazard ratio [HR], 0.66). Mutation analysis of HRR genes was possible in 97.7% of cases (778 cases) using samples derived from tissues or blood cell-free DNA. Of these, in 226 patients with HRR gene mutations, the median ibPFS in the combination group had not yet been reached, while it was 13.9 months in the placebo group. The difference was statistically significant, with an HR of 0.50. In contrast, in 552 patients without HRR gene mutations, median ibPFS in the combination group was 24.1 months while it was 19.0 months in the placebo group, and the difference was also statistically significant (HR, 0.76).

Overall survival, a secondary endpoint, was recently reported at the 2023 edition of the American Society of Clinical Oncology Genitourinary (ASCO-GU 2023) [55]. Median OS was 42.1 months for the olaparib (Lynparza^®^) plus abiraterone arm versus 34.7 months for the placebo plus abiraterone arm, representing a 7.4-month absolute difference in median OS compared to the standard care (at 47.9% maturity; HR, 0.81; 95% CI, 0.67–1.00; *p* = 0.0544). However, this trend was not statistically significant. Furthermore, theoretical background of the combined use of olaparib (Lynparza^®^) and abiraterone for ARSIs-naïve mCRPC is still weak, and there is a significant additional benefit of olaparib (Lynparza^®^) in cases with mutations in the HRR gene compared to cases without mutations [55]. Moreover, patients with *BRCA* mutations favored abiraterone and olaparib over abiraterone and placebo (HR, 0.29) compared to patients with non-*BRCA* mutations (HR, 0.91). In addition, the fact that the frequency of pulmonary infarction was higher in the combination therapy group than in the monotherapy group is a serious adverse event, and investigation of its cause is essential.

At the ASCO-GU 2023, an additional phase three trial focusing on a PARP inhibitor for mCRPC, TALAPRO-2, was reported [56]. This study evaluated the combination of a PARP inhibitor, talazoparib, and enzalutamide versus enzalutamide and placebo as first-line settings in patients with mCRPC without prior ARSIs. The study is unique in that it included two different cohorts of a genomically unselected (“all comers”) group (cohort 1), as well as an HRR gene mutated group (cohort 2). The primary endpoint of the study was rPFS and the secondary endpoints include OS, duration of response, PSA response of ≥90%, time to PSA progression, time to cytotoxic chemotherapy, time to antineoplastic therapy, PFS2 (PFS on the therapy that follows this study), and patient-reported outcomes (PROs). Furthermore, at a median follow-up of nearly 25 months in both arms, rPFS in a blinded independent review demonstrated a 37% reduction in the risk of progression or death in patients receiving talazoparib and enzalutamide [56]. However, median rPFS was not reached with the talazoparib and enzalutamide arm when compared to 21.9 months (95% CI, 16.6–25.1) in the placebo arm. When examined for HRR-mutated and HRR-unmutated groups, the benefit of talazoparib was clear in the HRR gene-mutated group, as anticipated (27.9 vs. 16.4 months; HR, 0.46; *p* < 0.001). In addition, treatment-emergent adverse events (TEAEs) with high rates of cytopenia in the experimental arm have been reported [56]. The most common TEAEs resulting from talazoparib dose reduction were anemia (43.2%), neutropenia (15.1%), and thrombocytopenia (5.5%) [56]. Similar to the PROpel study, we should wait for a summary of OS to determine whether the addition of talazoparib to enzalutamide in genomically unselected mCRPC patients would be favorable (Table 1).

Although the mechanism of resistance to PARP inhibitors in prostate cancer remains unknown, clinical trials are being conducted to confirm the effect of combining PARP inhibitors with other drugs. For example, *BRCA1*-deficient PARP inhibitor-resistant cell lines have been shown in vitro to rely on ATR for the repair of DSBs [57]. Based on this theoretical background, a phase two clinical trial of olaparib (Lynparza^®^) in combination with an ATR inhibitor is being conducted for mCRPC (NCT 03787680). Additionally, a phase three clinical trial (KEYLYNK-010) was conducted to assess the efficacy and safety of the combination of olaparib (Lynparza^®^) and pembrolizumab in the treatment of participants with mCRPC who have failed to respond to either abiraterone acetate or enzalutamide (but not both) and chemotherapy. However, the trial was discontinued due to unpromising results in the early interim analysis.

Therefore, we should wait for clinical trials based on the basic theoretical background and the results of meaningful clinical trials.

## 12. Conclusions

The DDR network is crucial for genome maintenance and is therefore essential for cell survival and proliferation [5]. Considering that these activities and the genes encoding DDR factors are frequently mutated in cancer, DDR could be a feasible target for suppressing the growth of cancer cells. PARP inhibitors have been established based on the concept of synthetic lethality in *BRCA1/2* mutant cancer cells, and they represent a great breakthrough in precision medicine for prostate cancer treatment [4]. However, PARP inhibitors are not effective in all patients with *BRCA1/2* mutant prostate cancer, and the inactivation of DDR genes elicits genomic instability, leading to alterations in multiple genes and eventually to drug resistance [34]. In this context, the use of PARP inhibitors and immune checkpoint blockades could be a promising strategy for a deeper understanding of tumor-microenvironment immunity.

## 13. Future Directions

Although technologies that enable the sequencing of patient samples have become easily available, analyzing and interpreting genomic and transcriptomic data to understand the mechanisms of disease occurrence, advancement, and drug resistance remain obstacles. It is imperative for us to concentrate our efforts towards understanding the disease mechanism and discovering novel targets for drug development as well as identifying markers for selecting appropriate patients, optimizing existing drugs through superior drug application, and identifying potent drug combinations that can maximize drug efficacy and reduce side effects.

## Figures and Tables

**Figure 1 cancers-15-02662-f001:**
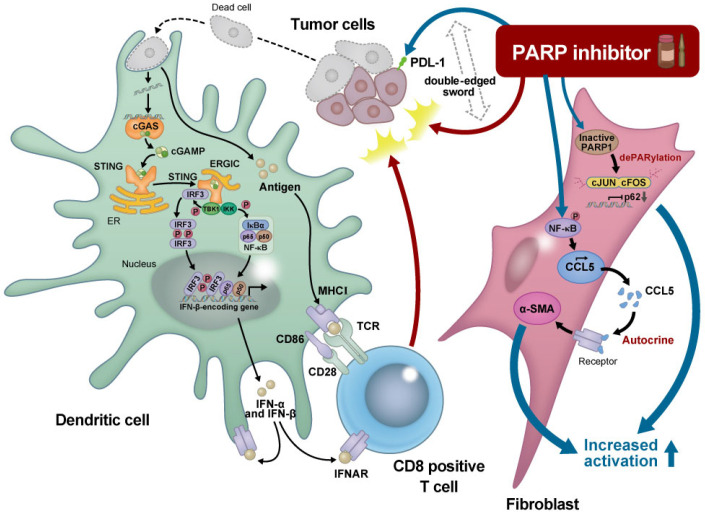
Schematic representation of the effect of a PARP inhibitor on the tumor microenvironment.

**Table 1 cancers-15-02662-t001:** Summary of representative clinical phase three trials on PARP inhibitors in prostate cancer.

Trial	PROfound				PROpel		TALAPRO2	
Treatment	Olaparib		Abiraterone/ Enzalutamide		Olaparib + Abiraterone	Placecbo + Abiraterone	Talazoparib + Enzalutamide	Placebo + Enzalutamide
Patient eligibility	mCRPC refractory against docetaxel and either abiraterone/enzalutamide				mCRPC without abiraterone and other second-generation AR inhibitor treatment (docetaxel was allowed as a neoadjuvant or adjuvant treatment after localized disease or as a first-line treatment for mHSPC)		mCRPC without second generation AR inhibitors treatment (enzalutamide, apalutamide, darolutamide); docetaxel and abiraterone was allowed as a treatment for mHSPC	
Genetic background	*BRCA1/2, ATM*	12 HRR-related genes other than *BRCA1/2, ATM*	*BRCA1/2, ATM*	12 HRR-related genes other than *BRCA1/2, ATM*	All comers		All comers	
Number of patients	162	94	83	48	399	397	402	403
Primary endpoint	rPFS of patients with *BRCA1/2* or *ATM* mutations				rPFS		rPFS	
	7.4 ms	-	3.6 ms	-	24.8 ms	16.6 ms	NR	21.9 ms
	HR, 0.34; (CI, 0.25–0.47) *p* < 0.001				HR, 0.66; (CI, 0.55–0.81) *p* < 0.0001		HR, 0.63; (CI, 0.51–0.78) *p* < 0.001	

AR: androgen receptor, mCRPC: metastatic castration-resistant prostate cancer, rPFS: radiological progression-free survival, mHSPC: metastatic hormone-sensitive prostate cancer, HR: hazard ratio, NR: not reach.

## Data Availability

No new data were created or analyzed in this study. Data sharing is not applicable to this article.

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
