# Peer review of "Roles of the PARP Inhibitor in BRCA1 and BRCA2 Pathogenic Mutated Metastatic Prostate Cancer: Direct Functions and Modification of the Tumor Microenvironment"

_cancers, 2023, doi:10.3390/cancers15092662_

Round 1
Reviewer 1 Report
The authors submitted a manuscript for review entitled, "Roles of the PARP inhibitor in BRCA1 and BRCA2 pathogenic mutated metastatic prostate cancer: direct functions and modification of the tumor microenvironment."
I would like to thank the authors for compiling very relevant and recent studies where the use of PARP inhibitors was presented in an organized manner. The mechanism of action for several inhibitors was discussed in detail, which indicates the author's research interests, but nevertheless, provided the reader with insight into the potential utility of these therapy modalities. The dynamic role of BRCA1/BRCA2 pathogenic variants was very important to discuss, especially since these therapies are used in clinical trials. If published, this review article will be a valuable resource for researchers in various fields where tumor death, therapy response, tumor metabolism and metastasis are relevant.
I don't have many criticisms of this manuscript, but the suggestions or changes to the grammar and presentation of the paper are listed below relative to the line number.
Line 11: add "(PARP)" as an acronym
Line 11: Change to "shown as an effective"
Line 12: Change to "therapeutic option for the disease."
Line 12: add "PARP" instead of the words spelled out
Line 13: Change to "cancer, and discuss"
Line 35: Omit "uniform medical or"
Line 38: Please restructure this sentence, it tends to run on and is confusing.
Line 42: Change is to "was"
Line 44: Omit by in the word "byproducts"
Line 89: Omit second comma after "Additionally"
Line 89: Omit "basically"
Line 96: Change the word "overseas" to "abroad"
Line 137: Add the frequencies of the genes HOXB13, ATM, BRCA1 because the frequencies were stated for BRCA2 in the previous sentence.
Line 186: Use the same format to cite references. Here there is a superscript for reference 26, but brackets at the end of the sentence for reference 27.
Line 236: Add "This adaption is sufficient"
English is my first language, and from what I read the manuscript was easy to understand. There are some idiosyncrasies and nuances to language, but the science came out and was very compelling. I appreciated that the authors wrote the review with several sections. After reading this review, I could use it as a conceptual resource and way to learn new methods to interrogate PARP and related signaling pathways.
Author Response
Response to the reviewer #1
Thank you for reviewing our article and giving us some comments. We have changed the manuscript according to the comments as below:
Line 11: add "(PARP)" as an acronym
>We have added “(PARP)”.
Line 11: Change to "shown as an effective"
>We have added “an”.
Line 12: Change to "therapeutic option for the disease."
>We have added “therapeutic”.
Line 12: add "PARP" instead of the words spelled out
>We have changed as the reviewer’s suggestion.
Line 13: Change to "cancer, and discuss"
>We have changed as the reviewer’s suggestion.
Line 35: Omit "uniform medical or"
>We have deleted “uniform”
Line 38: Please restructure this sentence, it tends to run on and is confusing.
>For clarity, we have changed the sentence as following:
Recently, it has become a standard treatment to use one of androgen receptor signaling inhibitors (ARSIs) or/and docetaxel in addition to medical or surgical castration from the initial treatment, according to the risk of prostate cancer.
Line 42: Change is to "was"
>We have changed as the reviewer’s suggestion.
Line 44: Omit by in the word "byproducts"
>We have changed as the reviewer’s suggestion.
Line 89: Omit second comma after "Additionally"
>Thank you for your suggestion. We have deleted it.
Line 89: Omit "basically"
>We have omitted it.
Line 96: Change the word "overseas" to "abroad"
>We have changed according to the suggestion.
Line 137: Add the frequencies of the genes HOXB13, ATM, BRCA1 because the frequencies were stated for BRCA2 in the previous sentence.
>Thank you for your thoughtful suggestion. We added some information as following:
Moreover, frequencies of other germline mutations were detected in the HOXB13 (0.8% in the prostate cancer cases and 0.2% in the control group), ATM (0.5% in the prostate cancer cases and 0.2% in the control group), and BRCA1 (0.2% in the prostate cancer cases and 0.1% in the control group) genes, in this order.
Line 186: Use the same format to cite references. Here there is a superscript for reference 26, but brackets at the end of the sentence for reference 27.
>We have revised according to the suggestion. Thank you.
Line 236: Add "This adaption is sufficient"
>We have added “adaptation” for clarity.
Reviewer 2 Report
This review article, authored by Takahiro Inoue, stands out as an exemplary piece of work. As evident from the abstract, the author has meticulously reviewed several related studies and summarized their findings in a comprehensive yet simple manner. In the era of precision medicine, a better understanding of basic research and clinical trial results is critical in paving the way for future research and policymaking in cancer management and insurance coverage.
Regarding the PARP inhibitors, particularly Olaparib, it would be helpful to know the differences between Olaparib and other PARP inhibitors, aside from their inhibition effects. What other factors could contribute to these differences? Also, are there any ongoing clinical trials on this topic?
The IMPACT study suggests that BRCA1 may not be a suitable indication for Olaparib treatment. However, in the PROfound study, BRCA1 was not separated from BRCA2. The same question applies to ATM and BRCA2. Could you please clarify the reason for this discrepancy?
Since Enzalutamide has shown favorable results in treating mCRPC, it would be intriguing to know whether Olaparib could provide more benefits to patients than Talazoparib did in the TALAPRO2 trial.
Finally, it would be beneficial to understand the rPFS results in patients with 12 HRR-related gene variants other than BRCA1/2 and ATM in the PROfound study, I wonder if it was mentioned in the trial?
Author Response
Response to the reviewer#2
Thank you for reviewing our article and giving us some comments. We have changed the manuscript according to the comments as below:
Regarding the PARP inhibitors, particularly Olaparib, it would be helpful to know the differences between Olaparib and other PARP inhibitors, aside from their inhibition effects. What other factors could contribute to these differences? Also, are there any ongoing clinical trials on this topic?
>Thank you for your thoughtful suggestion.
We have added some description as below, since niraparib showed highest tumor distribution.
“Tissue distribution and efficacy was also difference among these PARPi; niraparib showed highest tumor and bone marrow distribution (Synder 2022 PMID 35499386).”
There are no clinical trial comparing Olaparib vs any other PARP inhibitors against prostate cancer according to “Clinical Trial.gov.”
The IMPACT study suggests that BRCA1 may not be a suitable indication for Olaparib treatment. However, in the PROfound study, BRCA1 was not separated from BRCA2. The same question applies to ATM and BRCA2. Could you please clarify the reason for this discrepancy?
>The IMPACT study is evaluating targeted prostate cancer screening using PSA in men with germline BRCA1/2 mutations and is not evaluating treatment efficacy of Olaparib. Recent retrospective study showed that PARP inhibitor efficacy was diminished among patients with BRCA1 alterations rather than those with BRCA2 one. This is because the latter patients more frequently associated with monoallelic mutations and a lesser prevalence of concurrent TP53 alterations. We have already described briefly in session 8 using a reference (Taza, F.; Holler, A. E.; Fu, W.; Wang, H.; Adra, N.; Albany, C.; Ashkar, R.; Cheng, H. H.; Sokolova, A. O.; Agarwal, N.; et al. Differential Activity of PARP Inhibitors in. JCO Precis Oncol 2021, 5. DOI: 10.1200/PO.21.00070.). However, this study is a retrospective one, and we should need a further prospective study to confirm the result.
Number of prostate cancer patients with BRCA1 mutations is relatively small than that with BRCA2 (Interim Results from the IMPACT Study: Evidence for Prostate-specific Antigen Screening in BRCA2 Mutation Carriers. Eur Urol 2019, 76 (6), 831-842. DOI: 10.1016/j.eururo.2019.08.019.). Thus, prostate cancer patients with BRCA1 and BRCA2 might not be separated in the PROfoud study.
As for ATM mutations, we have already described briefly in session 8 using a reference
(Carreira, S.; Porta, N.; Arce-Gallego, S.; Seed, G.; Llop-Guevara, A.; Bianchini, D.; Rescigno, P.; Paschalis, A.; Bertan, C.; Baker, C.; et al. Biomarkers Associating with PARP Inhibitor Benefit in Prostate Cancer in the TOPARP-B Trial. Cancer Discov 2021, 11 (11), 2812-2827. DOI: 10.1158/2159-8290.CD-21-0007.).
Since Enzalutamide has shown favorable results in treating mCRPC, it would be intriguing to know whether Olaparib could provide more benefits to patients than Talazoparib did in the TALAPRO2 trial.
Finally, it would be beneficial to understand the rPFS results in patients with 12 HRR-related gene variants other than BRCA1/2 and ATM in the PROfound study, I wonder if it was mentioned in the trial?
>The rPFS results in patients with 12 HRR-related gene variants other than BRCA1/2 and ATM in the PROfound study can be found in the supplementary BRCA1/2 and ATM in the PROfound study data (Supplement to: de Bono J, Mateo J, Fizazi K, et al. Olaparib for metastatic castration-resistant prostate cancer. N Engl J Med 2020;382:2091-102. DOI: 10.1056/NEJMoa1911440).
Reviewer 3 Report
This is a well-written paper on the background of PARPi and its effect on tumor microenvironment in prostate cancer. What is most interesting to me is the difference in approvals in Japan vs the US. While the US has a range of HRR Mutations approved, in Japan only BRCA 1 and BRCA2 are approved, although ATM was also in cohort A. While Cohort B, had nonsignificant results, this was still approved in the US by combining Cohort A+Cohort B. I believe since both were based on the same trial, Profound but led to different mutation-based approval for Lynparza, I think this angle is very interesting to highlight to the readers. If the authors want to highlight this more in the title, abstract, and conclusions, it might be worth considering, although not necessary. One aspect that could help further improve this paper, is the addition of a table of ongoing trials in combination therapy that is currently ongoing and would be interesting to watch out for. Also if other new PARPi are in development, that would be interesting to highlight as well. Also examining HRD or homologous recombination deficiency is standard of care in ovarian cancer when considering PARPi, however, no research has been done in prostate cancer. If authors want to weigh in on this, they can, although this is a minor point and not necessary. The authors note an important point regarding molecular testing with liquid and tissue tests. It would be great if the authors could add what the guidelines recommend regarding the time of testing and what tests to use. Should the testing be done earlier or while the patient is progressing on AR-targeted therapy? Are both somatic and germline testing for BRCA recommended by guidelines in Japan? How does this differ from US guidelines? Only by testing, pts that are eligible for PARPi can be identified, therefore it would be important to add this. Thank you.English is good. A minor spell check is needed.
Author Response
Response to the reviewer#3
Thank you for reviewing our article and giving us some comments. We have changed the manuscript according to the comments as below:
>Thank you for your thoughtful comments. According to the reviewer’s suggestion, we added some description as below:
, if patients become refractory against an ARSI. [26]. The recommended timeline for a molecular test is different between Japan and US/European countries; in the latter countries it is recommend when diagnosing metastatic disease [3, 27].
Reviewer 4 Report
This is well-constructed and well-written review summarizing the current state of knowledge on the effect of PARP inhibitors in metastatic prostate cancer with pathogenic mutations in BRCA1 and BRCA2 genes. The topic of this review is timely and important because: (1) pathogenic BRCA1/BRCA2 variants (germline or somatic) are present in 4-18% of patients with localized/castration-resistant prostate cancer, (2) the review describes how PARP inhibitors can modify the tumor microenvironment.
The review is based on 55 well-chosen relevant articles published up to and including 2023.
In this review Authors:
- provide an overview of currently available and emerging PARP inhibitors for treatment of metastatic castration-resistant prostate cancer (mCRPC) with pathogenic variants in BRCA1 and BRCA2.
- analyze results of clinical trials of PARP inhibitors applied for mCRPC especially trials involving Olaparib.
- point out that PARP inhibitors have various functions, so in order to increase effectiveness of treatment with PARP inhibitors not only their role in DNA repair but also their manifold, multidirectional activity in modulating tumor microenvironment has to be understood. To this end, of special interest is the paragraph devoted to modulation of the tumor microenvironment by PARP inhibitors, together with fig.1 which makes clear the intricacies of the PARP inhibitor's effect on the tumor microenvironment. This paragraph and figure 1 make it clear that understanding the mechanisms underlying the modulation of the components of tumor microenvironment by PARP inhibitors will help to enhance their effects against tumors, including prostate cancer.
- describe important clinical and pathological characteristics of prostate cancer in patients with BRCA1/BRCA2 pathogenic variants.
- discuss mechanisms of resistance to PARP inhibitors.
I suggest the following small corrections:
- Ref. 3 is not appropriate. It deals with prostate cancer early detection, not treatment. It should be replaced by another ref. supporting this sentence.
- Line 105: 7 in superscript (... DNA replication7) should be printed in brackets or deleted. Line 111: the same with 15, line 186: same with 26, line 216 with 4, line 376 with 53.
In summary, this is well-written, needed, timely up-to-date review, covering all important aspects of application of PARP inhibitors in treatment of mCRPC and associated tumor-microenvironment immunity. It is a concise but comprehensive analysis of the relevant literature.
Author Response
Response to the reviewer#4
Thank you for reviewing our article and giving us some comments. We have changed the manuscript according to the comments as below:
- Ref. 3 is not appropriate. It deals with prostate cancer early detection, not treatment. It should be replaced by another ref. supporting this sentence.
>Thank you for your appropriate suggestion. We have changed to an appropriate reference.
- Line 105: 7 in superscript (... DNA replication7) should be printed in brackets or deleted. Line 111: the same with 15, line 186: same with 26, line 216 with 4, line 376 with 53.
>Thank you, we have changed accordingly.